# Long-Term Effects of ART on the Health of the Offspring

**DOI:** 10.3390/ijms241713564

**Published:** 2023-09-01

**Authors:** Hamid Ahmadi, Leili Aghebati-Maleki, Shima Rashidiani, Timea Csabai, Obodo Basil Nnaemeka, Julia Szekeres-Bartho

**Affiliations:** 1Department of Medical Biology and Central Electron Microscope Laboratory, Medical School, Pécs University, 7624 Pécs, Hungary; hamid_ahm1986@yahoo.com (H.A.); timi.csabai@gmail.com (T.C.); 2Department of Immunology, School of Medicine, Tabriz University of Medical Sciences, Tabriz 5165665931, Iran; leili_aghebati_maleki@yahoo.com; 3Immunology Research Center, Tabriz University of Medical Sciences, Tabriz 5165665931, Iran; 4Department of Medical Biochemistry, Medical School, Pécs University, 7624 Pécs, Hungary; rashidianishima@gmail.com; 5János Szentágothai Research Centre, Pécs University, 7624 Pécs, Hungary; 6Endocrine Studies, Centre of Excellence, Pécs University, 7624 Pécs, Hungary; 7National Laboratory of Human Reproduction, 7624 Pécs, Hungary; 8Department of Laboratory Diagnostics, Faculty of Health Sciences, Pécs University, 7621 Pécs, Hungary; basil.obodon@gmail.com; 9MTA—PTE Human Reproduction Research Group, 7624 Pecs, Hungary

**Keywords:** ART, stress, epigenetics, long-term effects

## Abstract

Assisted reproductive technologies (ART) significantly increase the chance of successful pregnancy and live birth in infertile couples. The different procedures for ART, including in vitro fertilization (IVF), intracytoplasmic sperm injection (ICSI), intrauterine insemination (IUI), and gamete intrafallopian tube transfer (GIFT), are widely used to overcome infertility-related problems. In spite of its inarguable usefulness, concerns about the health consequences of ART-conceived babies have been raised. There are reports about the association of ART with birth defects and health complications, e.g., malignancies, high blood pressure, generalized vascular functional disorders, asthma and metabolic disorders in later life. It has been suggested that hormonal treatment of the mother, and the artificial environment during the manipulation of gametes and embryos may cause genomic and epigenetic alterations and subsequent complications in the health status of ART-conceived babies. In the current study, we aimed to review the possible long-term consequences of different ART procedures on the subsequent health status of ART-conceived offspring, considering the confounding factors that might account for/contribute to the long-term consequences.

## 1. Introduction

Forty years after the birth of the first in vitro fertilization (IVF)- conceived baby and more than 27 years after the first intracytoplasmic sperm injection (ICSI), these techniques are widely used to overcome infertility-related problems [1]. It has been estimated that around 200,000 babies are annually born using assisted reproductive technologies (ART) worldwide, accounting for 5% of all conceived babies in developed countries [2]. In parallel with higher demands for ART, concerns about the health status of ART-conceived babies have also increased [3].

The concluding results from a careful analysis of epidemiological data confirmed that ART is associated with a slight increase in birth defects, with an anomaly rate of 3–4% at birth versus 2–3% at natural conception [4]. Although some studies found no association between ART and the health of the offspring, others suggested that ART-conceived babies exhibited a higher prevalence rate of cancer [5,6], high blood pressure, generalized vascular functional disorders, asthma, as well as metabolic and epigenetic disorders [7,8,9].

According to the theory of the developmental origin of health and disease, a suboptimal early-life environment increases the risk for non-communicable diseases throughout life. Pre-pregnancy, pre-implantation, and intrauterine environments might affect gene expression, adult phenotype, and susceptibility to disease in the offspring’s later life [10]. During ART processes, altered environmental conditions, including hormone administration, gametes, and embryo manipulation, and interfering in the natural selection of gametes in the preimplantation period lead to genomic and epigenetic alterations, which might result in enhanced incidence risk of malformations and disorders throughout the later life of ART-conceived babies [11]. Preimplantation embryos are highly sensitive to environmental factors. Some of the potential stress factors result in impaired embryo development and implantation failure, while others might have long-term effects.

However, the population that requires ART to conceive is different from fertile patients in many respects. Furthermore, infertility in itself might affect the offspring; the mothers are considerably older at the time of conception than the fertile population. Finally, yet importantly, many of the ART-conceived mothers had multiple pregnancies. Therefore, the available evidence must be handled carefully before reaching a conclusion [12]. The current study is aimed to review the data concerning the association between infertility treatment and offspring’s health status in later life.

## 2. Assisted Reproductive Technologies (ART)

The infertility treatment protocol should particularly target the diagnosed cause of infertility, applying proper methods that guarantee the highest chance for successful pregnancy and delivery of viable offspring. Infertility treatments include three main therapeutic strategies: pharmacologic therapy, surgical therapy, and ART. Recent decades have witnessed great development in ART, resulting in the treatment of previously untreatable and challenging cases [8]. Since the birth of Louis Brown in 1978, over 8 million babies have been born using different ART procedures [13].

Over the past four decades, several important new techniques have been introduced. Adaptation of ultrasonography to evaluate ovarian follicles, the introduction of controlled ovarian hyperstimulation with aromatase inhibitors, clomiphene citrate and menopausal gonadotrophins, administration of gonadotropin-releasing hormone (GnRH) agonist and antagonist in stimulation protocols, performing oocyte retrieval under ultrasonographic guidance, embryo freezing, and ultra-rapid embryo freezing, have resulted in increased efficiency of ART [14,15,16,17].

The first successful pregnancy and live birth after microsurgical epididymal sperm aspiration (MESA) was reported in 1988, followed by live birth after intracytoplasmic sperm injection (ICSI) in 1992. These procedures are highly effective in overcoming male factor infertility and are considered significant progress in the ART world [18,19].

IVF, ICSI, and, in the past, gamete intrafallopian tube transfer (GIFT) and zygote intrafallopian tube transfer (ZIFT) are/were the procedures to treat infertility in humans. In IVF, fertilization occurs as a result of adding a defined amount of sperm to the extracted oocyte in a culture medium, followed by culture and transfer of the embryo into the uterus, whereas in ICSI a single spermatozoon is injected into the cytoplasm of a mature oocyte [18]. ICSI is associated with high fertilization and pregnancy rates regardless of sperm concentration, morphology, and motility in male-related infertility; on the other hand, the competition among the spermatozoa is eliminated. GIFT could be used as an alternative to IVF in women with at least one patent tube. Intrauterine insemination (IUI) is often the first therapeutic intervention that is offered to treat subfertility. In this procedure, the sperm are brought closer to the oocyte for fertilization at the appropriate time. Economic evidence favors an IUI trial of three to four cycles prior to IVF in many cases and provides for in vivo embryo development [18,19].

In vitro maturation (IVM) of immature oocytes was another important step in ART development [20]. In IVM, immature follicles are minimally exposed to hormonal stimulation, therefore this is an alternative approach for IVF in women with polycystic ovary syndrome (PCOS), who are at risk of ovarian hyperstimulation syndrome.

Recombinant gonadotrophins have been used in hyperstimulation protocols since 1992 [21]. Preimplantation genetic testing (PGT) was introduced in 1999 and the first live birth after blastocyst biopsy was reported three years later [22,23]. Figure 1 shows the different ART procedures.

## 3. Preimplantation Stress and Development

David Barker was the first who clearly articulated the developmental origin of health and disease theory [24]. During development, there is a specific period when the organism is plastic and sensitive to its surrounding environment. While plasticity provides adaptation to the environment and functions as a survival mechanism for the developing embryo, environmental stimuli during early development generate permanent effects that persist throughout the lifespan [24,25] and predispose the resultant offspring to diseases in adulthood [26,27].

During the fertilization and preimplantation period, organisms are highly vulnerable to environmental conditions [28]. Preimplantation development is characterized by highly coordinated physiological and epigenetic alterations as the zygote develops into a blastocyst. The diverse energy requirement for this transitional phase are supplied by the spatio-temporal availability of nutrients, oxygen, and growth factors as the developing embryo travels from the oviduct to the uterus.

It is accepted that the intrauterine environment affects gene expression, adult phenotypes, and disease susceptibility. Animal experiments confirmed the effect of varying pre-implantation and prenatal conditions on the offspring phenotype. Feeding rats with a low-protein diet for 4 days prior to implantation resulted in lower blastocyst cell numbers lower perinatal and postnatal growth rates, and increased incidence of cardiovascular disease, hypertension, and mild hyperglycemia in adult offspring [26,29]. Maternal malnutrition has similar effects on the offspring in humans [25] In another study, the low-protein diet of the mother caused adult hypertension, as well as hyperactive behavior and altered postnatal growth in mice [27]. Adamiak et al. showed a correlation between undernutrition during the prenatal period and increased vasoconstriction, as well as impaired glucose tolerance in offspring in cattle [30].

Preimplantation development encompasses the fertilization of the oocyte by the sperm through the implantation of the hatched blastocyst into the uterus lining. Following the fertilization of the oocyte, the single-cell zygote is formed, which undergoes several rounds of cleavage without increasing the whole embryo volume [31]. As the embryo reaches the eight-cell stage, polarization allows compaction, which provides a foundation for establishing distinct cell lineages: the trophectoderm (TE) and inner cell mass (ICM). These cell lineages evolve over subsequent asymmetric cleavage divisions, continuing through the formation of the blastocoel cavity. Cavitation results in the full expansion of the blastocyst, which hatches from its surrounding zona pellucid layer before implanting into the endometrium [32].

These stages are characterized by different nutritional requirements. During the progression from the zygote to the two-cell stage, there is an absolute requirement for pyruvate, while further developmental stages are highly supported by lactate [33].

Glucose is not required until the late four-cell/early eight-cell stage but becomes the main carbohydrate metabolized during embryo compaction [34]. The transition in metabolic and energy substrate requirements is affected by the availability of amino acids, carbohydrates, and extracellular pH, which influences oxidative phosphorylation, glycolytic activity, and membrane transport, with potential effects on proliferation and growth [35,36]. The development of the preimplantation embryo occurs over a dynamic spectrum of conditions, indicating strong metabolic plasticity and the capacity of the embryo to compensate for metabolic fluctuations at this time [37,38].

During mammalian preimplantation development, coordinated reprogramming of the genome occurs, which maintains after birth [39]. Since preimplantation development is characterized by epigenetic rearrangement, the embryos may be particularly vulnerable to insults. Environmental changes during the preimplantation period could cause errors or epimutations or alter the programming of the cell states and response pathways [40].

Evidence from animal studies suggest that stress limited to this period can exert both short and long-term effects [27,30]. In addition to metabolic conditions, toxicologic agents can affect early pregnancy [41]. Developmental exposure to diethylstilbestrol or bisphenol A activates estrogen-responsive genes, affects proliferation, and increases susceptibility to cancer in adulthood [41]. Higher doses of bisphenol A disrupted embryo growth and development in mice [42]. Smoking before pregnancy impairs embryonic development. Tobacco exposure is associated with decreased ovulation rate, altered tubal function, and perturbated embryo growth in female mice [43]. In mice, tobacco exposure reduced both sperm count and preimplantation embryo development [44].

During the laboratory manipulation of ART, the gametes and embryos are exposed to different kinds of stress, e.g., in vitro culture of the embryos, light, etc. During natural conception, mammalian gametes and embryos are not exposed to direct light and therefore they may not have developed the protective mechanisms [45]. The laboratory manipulation of light-related damage to oocytes and embryos can be seen at various levels, including mitochondrial degradation [46], DNA fragmentation [47] and ROS production in the cytoplasm [48]. Bognar et al. showed that exposure to white light impairs the implantation potential of in vitro cultured mouse embryos. The harmful effects, which were related to the wavelength rather than to the brightness of the light, could be partly corrected by using a red filter [49]. Based on these results, in a human IVF facility, red filters were applied on laboratory lamps and UV or infrared filters in microscopes in order to eliminate white light exposure of the cells throughout all work stages. These precautions resulted in significantly higher fertilization, blastocyst development, and better clinical pregnancy rates in light-protected than in non-protected ICSI cycles [50].

Different types of preimplantation stresses and their effects on embryo development and long-term effects are summarized in Table 1.

Animal models are essential to evaluate the effect of preimplantation stress on subsequent development and health status and to understand the mechanisms underlying the embryo stress response and potential outcomes later in life [40]. Since animal studies remove infertility as a potentially confounding factor, they allow the assessment of a particular controlled stress and extend investigations with available molecular tools. Such studies may provide useful information that could help to authenticate human ART findings [69,70]. However, animal studies are not completely immune to errors and bias [71].

It is documented that in vitro culture postpones embryo development by 18 to 24 h [72]. Variables such as the composition of culture media, fertilization procedures, oxygen tension, temperature fluctuations, and culture dish stiffness can impact the ratio of the embryo developing to blastocyst, TE and ICM cell number and lineage ratio, embryo morphology and gene expression [11,56]. Schwarzer and colleagues investigated the effect of 13 different culture protocols on mouse embryo development and found that different culture conditions exerted distinct effects on developmental competence, lineage composition, and global gene expression [73]. Each of these differences could significantly impact offspring growth and health in later life. IVF and ICSI as well as in vitro culture reduce differential gene expression between inner cell mass and trophectoderm, downregulate the expression of genes involved in transportation of micronutrients and placentation as well as altered expression of various imprinted genes [52,54,56]. This is particularly relevant, since imprinted genes take part in regulating placentation, placenta efficiency, and nutrient transport capacity [74].

Fetuses from IVF have smaller sizes throughout the first half of gestation, but exhibit rapid catch-up growth to controls during the second half of pregnancy, probably due to enlarged placentae. At birth, there is no statistically significant difference in weight between IVF and control animals, suggesting that birth weight cannot be a strong indicator of fetal growth and nutrition [75].

Ecker and colleagues were the first who reported that stress during the preimplantation period can produce clear definable postnatal phenotypes. They demonstrated that culturing embryos from the two-cell to blastocyst stage results in adult male offspring with decreased anxiety and impaired spatial memory. The results also demonstrated that the culture or embryo transfer has no significant effects on embryo development to term [28]. Another study by Fernandez-Gonzalez et al. observed similar behavioral changes in mice transferred at the two-cell stage following ICSI [76].

Increased body weight after preimplantation culture, ICSI, or IVF has been described by several research groups [12,76,77]. IVF and ICSI-conceived mice exhibit signs of glucose intolerance and insulin resistance [77]. It has been indicated that in vitro stress affects organ size and integrity: in a study by Gonzalez et al., it was demonstrated that mice embryo culture is associated with kidney inflammation, pneumonia, and testicular atrophy, as well as possible liver steatosis, reflecting disrupted glycogen metabolism and subsequent lipid mobilization as a compensatory energy source [12]. It has also been suggested that the stress during embryo culture results in altered expression of several metabolic and cardiovascular genes, as well as increased systolic blood pressure in adult mice [78].

## 4. ART and Imprinting Disorders (IDs)

Epigenetics is defined as the study of inheritable alterations in gene function that do not entail a change in DNA sequence. Epigenetic modifications include adjustments to DNA bases, histone modification, post-translational regulations, and genomic imprinting [79]. It has been confirmed by human and animal studies that reprogramming of specific genes is important for proper embryo development [80,81].

Several studies suggest an association of ART with imprinting disorders. Rare genetic diseases, e.g., Beckwith–Weidemann syndrome, Angelman syndrome, Silver–Russell syndrome, and Prader–Willi syndrome seem to be more frequent among ART-conceived babies [82,83,84,85,86], but when comparing 75,000 ART conceived children to 2,775,239 naturally conceived children, only the incidence of Beckwith–Wiedemann syndrome was higher in the former group [81].

Though it is plausible that ART possibly interferes with the establishment and/or maintenance of imprints, the data are controversial and are not clear enough to allow conclusions. The patients’ data are incomplete and lack molecular characterizations of the diagnosis; furthermore, variables such as maternal age, infertility, underlying causes, and specific ART methods are often not included in the analyses [87].

ART procedures occur during a critically coordinated period of epigenetic reprogramming, the preimplantation period, which is highly vulnerable to epigenetic aberrations [88].

Because of the heterogeneous nature of ART procedures, differences between imprinted regions, and disparity in tissue samples and detection techniques, our understanding of the effects of ART on imprinting and DNA methylation is limited [89]. ART involves several processes, such as hormonal ovarian hyperstimulation, in vitro oocyte maturation, male gamete processing, methods of fertilization, cryopreservation of gametes and embryos, PGT, and embryo transfer, all of which can potentially disturb normal genomic imprinting [88].

Although it is not clear which aspects of the ART procedures are responsible for disturbed imprinting in humans, information has been obtained from studying mouse models. Several studies demonstrated DNA methylation errors in oocytes, embryos, and placentas following induced superovulation in mice [90,91,92,93]. Chen et al. reported that superovulation (but not IVF or IVM) had a significant effect on the mRNA expression and methylation level of paternally imprinted growth factor receptor-bound protein 10 (Grb10) and H19 (the gene regulating the production of noncoding RNAs) [91]. Grb10 regulates fetal and placental development, while after birth its expression is associated with the pathogenesis of Type 2 diabetes and Alzheimer’s disease [94,95].

Other studies confirmed the negative effects of hyperstimulation on imprinted genes H19, insulin-like growth factor 2 (Igf2), and small nuclear ribonucleoprotein polypeptide N (Snrpn) in mouse blastocysts [96,97]. The Igf2-H19 locus encodes critical paternally imprinted genes that regulate normal embryonic development. The removal of hypomethylation of the Igf2-H19 locus plays a significant role in orchestrating quiescence of pluripotent stem cells in adult organisms, and is probably involved in the regulation of lifespan [98].

Other components of ART procedures have been investigated in human and animal models. Altered methylation at KvDMRI (CpG island) was confirmed in the placenta and fetuses from cryopreserved mouse embryos [99].

### 4.1. Effects of Cryopreservation and Vitrification on Epigenetic Alteration

Gamete and embryo cryopreservation have become important techniques in ART facilities. In this process, embryos in a high concentration of cryoprotectant are plunged into liquid nitrogen. Despite the significant progress in vitrification techniques, there are still concerns regarding their short and long-term effects on embryos and the resulting offspring [100,101]. It has been demonstrated that cryopreservation is associated with increased DNA fragmentation and apoptotic gene expression [102], as well as a decreased rate of blastocyst hatching in mouse embryos [103]. Epigenetic disorders have also been shown in human and animal models as a result of using cryopreservation techniques. Chen et al. showed that oocyte vitrification causes epigenetic instabilities in bovine embryos. DNA methylation and H3K9me3 levels in oocytes and early cleavage embryos were lower, while the levels of acH3K9 were higher in vitrified embryos than in the controls [104]. While mouse embryo vitrification from the two-cell or eight-cell to blastocyst stages may result in reduced levels of global DNA methylation, vitrification seems to have no significant effects on the DNA methylation status of H19/IGF2 differentially methylation region (DMR) in human blastocysts [105]. Studies on human placentas from frozen-thawed embryo transfers (FET) have identified imprinting disorders that may be associated with the positive regulation of gene expression, growth, development, cell migration, and type II diabetes mellitus [106].

### 4.2. Effects of Culture Media on Epigenetic Alteration

The culture medium provides an environment for the developing embryo after fertilization, when much of the genome is demethylated. The methylation loss of imprinted genes is believed to result in fetal abnormalities [107,108]. Several mechanisms have been postulated for the aberrant imprinting that occurred in cultured embryos, such as alterations in the expression and subcellular localization of DNA methyltransferases (DNMTs) that are essential for imprint maintenance [84].

In vitro culture is probably the most relevant factor in the alterations of epigenetic reprogramming and development of embryos produced by ART. The importance of culture media in epigenetic preimplantation reprogramming and its impact on early embryo development has been confirmed by an increasing number of studies [109,110,111]. The suboptimal conditions of IVC exert effects on trophoblast development [112]. In mice, the disrupted epigenetic profile of trophoblast is maintained in the placenta, which seems to be more sensitive than the embryo to the IVC environment [113].

Epigenetic modifications in the preimplantation period have been linked to unbalanced fetal–placental development, abnormal fetal growth, and altered metabolic responses. Some studies have confirmed these findings, suggesting cellular aberrations in the placenta and fetus are connected to alterations in gene expression and the association of epigenetic disruption with glucose metabolism and fetal growth in mice [114,115].

Culture medium supplemented by serum has been linked to abnormal skeleton and organ development [116]. Mouse offspring from IVC embryos exhibit anxiety, special memory, and psychomotor activity alterations. The type of culture medium also has a certain impact on the extent of these behavioral changes [28].

### 4.3. Effects of In Vitro Maturation (IVM) on Epigenetic Alteration

IVM of oocytes is associated with genetic alterations [117]. In vitro matured germline vesicle oocytes show methylation gain at the H19 locus and a methylation loss at Igf2R and Mest/Peg1 locus compared to those obtained from the ovary [118]. In pigs, IVM oocytes have a decreased epigenetic competence and a reduced ability to transform sperm chromatin into the male pronucleus. Reduced active demethylation and histone H4 acetylation epigenetic competence in the male pronucleus (PN) in IVM zygotes has been demonstrated through the analysis of global DNA methylation in the late PN stage [119]. An altered methylation pattern of the H19 differentially methylated regions, which are normally unmethylated in maternal alleles, was detected in 5 out of 20 (25%) human IVM oocytes [120]. In another human study, one paternally-imprinted (GTL2) and three maternally-methylated (LIT1, SNRPN, and PEG3) imprinted genes were analyzed in 71 in vitro- and 38 in vivo matured oocytes. In vitro matured oocytes were retrieved from subjects with PCOS in minimally stimulated cycles without priming with human chorionic gonadotropin (hCG). Single-cell methylation analysis (using limiting dilution bisulfite sequencing) demonstrated no significant increase in imprinting mutations at GTL2 and LIT1, SNRPN, and PEG3 in IVM oocytes [121]. Pluisch and colleagues examined the possible transmission of epigenetic defects to the next generation. They measured some developmentally important genes and interspersed repeats in 11 human IVM offspring and 19 conventional ART-conceived counterparts as control. The results showed that IVM did not exert any significant effects on the chorionic villus and cord blood DNA methylation [122].

### 4.4. Effects of ICSI on Epigenetic Alteration

Studies on different animal models conceived by ICSI have shown an asynchronous remodeling of chromatin decondensation of the male pronucleus in primates, cattle, and mice [123,124,125]. ICSI-conceived mice compared to conventional IVF-conceived counterparts present long-lasting transcriptome disturbances that are maintained until the neonatal stage. However, these alterations have not been associated with changes in the phenotypic profile or with transgenerational effects [126]. Mouse blastocysts from ICSI have a reduced number of inner mass cells and significant differences in gene expression related to cell function, development, and metabolism [52]. These discrepancies are a result of the different strategies used to activate the oocyte and guarantee embryo development and the protocols used for ICSI in different species [127].

## 5. ART and Genomic Imprinting in the Placenta

The placenta regulates the growth and development of the embryo during gestation by establishing a bilateral connection, to control the exchange of nutrients and gas between the mother and fetus [128]. A series of imprinted genes are highly expressed in the placenta and play critical roles in its development. These genes take part in the growth, morphology, and nutrient transport capacity of the placenta, which control the nutrient supply for fetal growth [129]. Studies on transgenic mouse models have confirmed roles for imprinted genes in placental function and fetal development. It has been demonstrated that Ascl2, Phlda2, and Peg10 are important for proper placental morphology and function [130]. Igf2 has been indicated to take part in nutrient control, placental size, and morphology [130].

Studies evaluating the effects of ART on placenta development in humans have demonstrated that alterations in DNA methylation may result in dysregulated gene expression. Sakian and colleagues examined human ART-derived placentas and reported alterations in H19 and Igf2 gene expression, due to a loss of imprinting on the paternal allele [131]. It has also been indicated that, compared to naturally conceived human pregnancies, ART-derived placentas show an increased expression of MEST and SERPINF1 genes, a decreased gene expression of COPG2 and NNT genes, a loss of methylation in GRB10, MEST, PEG3, SERPINF1, SCL22A2, and no alterations in the methylation or expression of DLK1, GNAS, H19, and KCNQ10T1. Since these genes play roles in the development and differentiation of adipocytes, insulin signaling, and obesity, altered expression of these genes in the placenta could suggest an increased risk of metabolic syndrome in adulthood [132]. Shi et. al. reported an altered H19/Igf2 ICR methylation pattern in placentas from three IVF-conceived pregnancies. Nonetheless, these children appeared healthy. It was hypothesized that the methylation disruption was due to imprinting errors in the gametes or errors that occur during embryo culture [133]. Nelissen and colleagues observed a reduced DNA methylation in H19 and PEG1 and enhanced expression of H19 and PHLDA2 in human placentas from IVF/ICSI pregnancies, but they did not observe significant differences in body weight in the studied children [134].

Although some of these changes may be due to ART procedures, it is possible that these epigenetic alterations may be a result of the underlying infertility of the couples. Gamete’s methylation pattern can be altered prior to fertilization, which can influence gene expression, exerting effects on offspring phenotype [135,136]. Since human ART studies are affected by confounding factors, conducting studies on animal models has proven to be critical in determining the effects of ART. The results from animal studies have demonstrated that embryo culture and ART procedures can alter epigenetic gene regulation and cause placental and fetal abnormalities. In large animal studies, cattle and sheep, derived from in vitro embryo culture, often present with overgrowth syndrome and sheep with large offspring syndrome had abnormal methylation and reduced expression of the Igf2r gene [137].

Mouse models used to examine the effect of ART on placentation and development revealed that superovulation and IVF as well as oxygen exposure during embryo culture, can induce morphological and epigenetic alterations in the placenta by disturbing the methylation of the ICR of the H19/Igf2, SNRPN, PEG3, and KCNQ10T1 loci [138,139]. Taken together, optimizing ART procedures may decrease adverse effects, by minimizing the occurrence of placental abnormalities and epigenetic alterations.

## 6. ART and Underlying Parental Characteristics

Adverse perinatal outcomes associated with ART are related to the subfertility of couples, multiple pregnancies, and ART procedures [140]. It is well-documented that multiple pregnancies following multiple embryo transfers are the major cause of preterm delivery and low birth weight, with long-term health risks [141]. In addition, compared to spontaneous conception, singleton pregnancies from ART are more often associated with adverse obstetric and perinatal outcomes [142,143]. These include antepartum hemorrhage, low birth weight, preterm birth, stillbirth, small for gestational age babies, perinatal mortality, need for neonatal intensive care, pregnancy-related hypertensive disorders, preterm rupture of membranes, gestational diabetes, labor induction and Cesarean section [142,143,144,145,146,147].

It is not clear whether the ART procedures or the underlying parental characteristics or genetics are the main cause of the worse obstetric and perinatal outcomes. A large study by Seggers and colleagues using siblingship analysis showed that maternal characteristics, such as age and subfertility, but not IVF treatment, are linked with lower birth weight in IVF-conceived babies [148]. A systematic review and meta-analysis suggested that the risk of preterm birth and small for gestational age babies after IVF is higher in women with endometriosis compared to women without endometriosis [149]. A cohort study reported a higher risk for preterm birth and large for gestational age babies among IVF-conceived newborns from women with PCOS [150]. However, other studies indicated that women with unexplained infertility did not show a higher risk of adverse birth outcomes after IVF [146,151,152].

Huiting Yu et al. demonstrated that 9480 IVF-conceived singletons were at higher risk for adverse birth outcomes compared to singletons in the general population of Shanghai. Underlying factors of infertility, such as endometriosis, PCOS, semen abnormalities, and uterine factors, were linked to preterm birth and abnormal birth weight [153]. Another study demonstrated that male-factor infertility is correlated with low birth weight (LBW) at full term in ART-conceived singletons [154]. Uterine factors, such as uterine inflammation, anatomical defects, impaired receptivity, and cervical insufficiency, have been shown to be associated with an increased risk of preterm birth and low birthweight [155,156]. Some studies have reported that tubal factor infertility is associated with a higher risk of miscarriage, perinatal outcome preterm birth (PTB), and low birth weight (LBW) for singletons conceived by ART; the etiological reasons are mainly attributed to infections and inflammation [157,158,159].

The association between male factor infertility and increased risk of birth defects has not been completely determined. It has been demonstrated that semen abnormality is associated with a higher incidence of LBW, but not PTB. It has also been suggested that sperm abnormality, asthenozoospermia, and oligozoospermia are associated with an increased risk of LBW [153]. On the contrary, most other studies found no significant association between infertility and a higher risk of PTB and LBW compared to unexplained infertility [157]. Moreover, semen parameters did not affect embryo quality or live birth outcomes [160]. These results may partly be interpreted by ART-related aspects, such as improved endometrial receptivity, embryos with higher quality surviving the freezing–thawing process, and the effect of cryoprotectants [137,161]. ART procedures, such as in vitro culture, gametes and embryo manipulation, and freezing and thawing procedures during the preimplantation period can cause epigenetic alteration resulting in birth defects. Currently, the underlying mechanism is not fully understood, and epigenetic effects in ART-conceived offspring require further investigation. While perinatal outcomes, such as LGA and macrosomia, may not be a serious threat to infant survival, they are correlated with an increased risk of cardiovascular disease in adulthood [162,163]. Thus, the association between ART and perinatal outcome should be carefully investigated and longer-term follow-up of children’s development is also important.

## 7. ART and Congenital Malformations

IVF-conceived infants have an increased risk for neural tube defects, omphalocele, hypospadias, and alimentary tract artesia [164]. The increased risk of birth defects after ART may be associated with ovarian hyperstimulation [165] IVF or ICSI, in vitro culture of gametes and embryos, possible damage during preimplantation genetic testing (PGT), freezing–thawing, luteal support, and embryo transfer [166,167,168]. Some studies have shown a higher rate of malformations after ICSI compared to IVF [169,170], whereas others did not [171]. Luke et al., reported that the rate of major chromosomal birth defects in ART-conceived infants is 18% higher than in naturally conceived counterparts [172].

Some studies have reported a significantly increased risk of birth defects among ART-conceived babies [146,173,174,175,176].

The maternal age of ART patients is considerably higher than that of fertile women, and consequently subfertility in itself might account for the increased risk of birth defects [177]. Therefore, it cannot be ruled out that the underlying infertility, and not the ART procedures, are responsible for the higher rates of birth defects [151,178].

## 8. ART and Neurological Disorders

Different ART procedures could have different effects on neurodevelopment. La Rovere at al. reviewed the possible effects of epigenetic alterations (due to hormone exposure, gametes preparation, gametes, and embryo freezing, use of culture media, growth conditions for embryos, etc.) on the neurological development of the fetus [179].

Animal studies suggest the possibility of long-term and transgenerational effects of ART on the behaviour and neurodevelopment of adult mice [28]. Adult mice derived from in vitro cultured embryos showed hippocampus and behavioral alterations [28]. H19 mRNA expression was slightly decreased in blastocysts cultured in fetal calf serum supplemented medium, and the mice from these embryos showed anxiety and memory deficiencies at adulthood [12]. Blastomere biopsy results in impaired spatial learning, enhanced neuron degeneration, and altered expression of proteins involved in neural degeneration in aged mice in comparison to aged control mice [180]. These studies suggest that ART might affect the development of the nervous system, but the data from these studies are often inconsistent. A follow up to 12 years demonstrated an increased risk for cerebral palsy (CP) in ART-conceived children [181]. These results were confirmed by Lindegaard et. al., reporting an 80% increased risk of CP in children from IVF [182]. Subsequently, other authors, while confirming the increased risk of CP in children from ART, demonstrated that, following adjustment for confounding factors such as maternal age, multiple pregnancies, and preterm birth, this association became less evident [183,184]. Zhu et. al. demonstrated a higher risk of CP in ART-conceived children even after adjustment for multiple pregnancies and preterm delivery [185]. A more careful analysis of potential factors associated with both IVF and CP and following adjustments for maternal age, year of birth, parity, and smoking, concluded that ART is associated with only a moderately increased risk for CP, and even this might result from increased neonatal morbidity associated with multiple births [183].

Several studies have addressed the possible risk of intellectual disability in ART-conceived babies. Minimally increased risk for intellectual disability was reported, when gestational age, birth weight, socioeconomic and parental educational status were all taken in account [186,187,188,189].

As for the association between ART and increased risk of autism or autism spectrum disorder (ASD), the results are conflicting. Maimburg et. al. demonstrated a lower risk for infantile autism among ART-conceived children, even after adjusting risk factors related to ART and infantile autism [190]. Another study reported a slightly increased rate of ASD among ART-conceived children, which disappeared following the adjustment for confounding factors such as maternal age, parity, educational level, smoking, multiplicity, and birth weight [191]. Kissin and colleagues found a higher incidence of ASD during the first five years of life in ICSI-conceived children compared to IVF-conceived counterparts [192]. A meta-analysis including three cohort studies and eight case-control studies did not show an increased risk of ASD in ART-conceived children [193].

## 9. ART and Cancer Risk

A systematic meta-analysis by Hargreaves et al. revealed an increased overall cancer risk in children born after infertility treatment. The results showed that the offspring of women with fertility problems had a significantly increased risk for leukemia in childhood and for cancer of the endocrine glands in young adulthood [194]. In a meta-analysis by Chiavarini et al. the possible association between ART and childhood cancer was investigated, and an increased risk of all cancers in ART-conceived children was found [195] In particular, an increased risk of hematological cancers, leukemias, sarcomas, and hepatic cancers was observed in ART-conceived children. In contrast, no significant association between neuroblastoma, retinoblastoma, and solid tumors was reported [195]. Similarly, another meta-analysis reported an increased risk of all cancers, leukemias, hematological cancers, and neural cancers for ART-conceived children [194]. The results by Wang et. al. demonstrated that children conceived by ART had a significantly increased risk of all cancers, leukemia, hematological malignancies, and hepatic tumors [196].

The results indicating increased risk for specific cancer types are inconsistent across studies [197,198,199]. These studies had limitations, such as short follow-up duration, small sample size, restriction to a general population comparison group, and lack of adjustment for confounding factors. It has been shown by a large-scale study with 21 years of follow-up that overall cancer risk in ART-conceived children is not higher, neither when compared with naturally conceived babies born to sub-fertile women, nor when compared with the general population [200]. In a British study among 12,137 ART-conceived children with 7.9 years of follow-up, the overall cancer risk was not higher compared with the general population [197]. In a Swedish cohort study by Kallen et. al., no increased risk for cancer was observed among ICSI-conceived babies [201].

Studies that investigated the association between ART and cancer risks have several limitations. Since cancer in children and young adults is rare, the number was rather low among subgroups, despite the large sample size and long follow-up of the cohorts. Therefore, the reported not significantly increased risks must be interpreted carefully. Confounding factors, such as parental subfertility, child’s birth year, medical record information about ovulation induction, intrauterine insemination, and using expert knowledge about the clinical practice at the time, should be considered when the association of ART with cancer risk among ART-conceived children is investigated.

## 10. ART and Cardio Metabolic Disorders

Early epigenetic alterations resulting from ART may lead to adverse perinatal outcomes and chronic cardiometabolic disease in adulthood [202,203,204]. Endothelial dysfunction and increased stiffness of vasculature [205] and higher systolic blood pressure [206], as well as disrupted functions of fatty-acid metabolism-related enzymes [77] and altered glucose parameters [114,207], have been reported in ART-conceived mice. Human studies have also demonstrated elevated blood pressure and impaired vascular function [208,209], abnormal retinal vessel morphology, congenital heart defects and altered protein expression profile in the umbilical veins of ART-conceived children [210,211]. Furthermore, ART pregnancies are often associated with preterm birth and low birth weight babies, and it has been shown that small for gestational age babies were at increased risk of ischemic heart, and infants who are born preterm or with very low birth weight have modestly higher systolic blood pressure later in life [212,213]. A systematic review and meta-analysis by Gou et. al. included data from 19 studies (both singleton and twin pregnancies included) that investigated blood pressure, cardiovascular function, adiposity, and metabolism during childhood, puberty, and early adulthood [214]. The results showed a slightly significantly higher blood pressure, increased vessel thickness, and suboptimal cardiac diastolic function in ART-conceived offspring [214].

It has been demonstrated that children conceived by ART showed differences from controls in systemic circulation, artery structure, and systolic pulmonary artery pressure in a hypoxic situation [215]. In line with this study, Chen et. al. showed that IVF-conceived offspring develop increased blood pressure when fed with a high-caloric diet [216]. Ceelen et. al. reported increased glucose levels in pubertal IVF-conceived offspring independently of parental characteristics (subfertility, body weight, and maternal age) or early life factors (gestational age and birth weight) [217]. Chen and colleagues demonstrated only reduced peripheral insulin sensitivity in young adults from IVF. When the offspring were challenged with a high–caloric diet, they showed high fasting blood sugar, glucose intolerance, and insulin resistance [218]. However, other studies found no significant alterations in glucose metabolism after IVF [217,218]. Moreover, it has been shown that the ICSI procedure amplifies the sexual dimorphism in body fat accumulation and distribution at puberty (increased central adipose tissue in female offspring, and an increase in peripheral adiposity at pubertal age in males) [219]. Increased levels of triglycerides have been reported in ART-conceived offspring [220]. These studies show that ART procedures could impact lipid and glucose metabolism in the offspring, resulting in a high risk of type 2 diabetes and metabolic syndrome. Furthermore, the alterations appear to be more evident at pubertal age rather than in early childhood, indicating that even if ART-conceived offspring seems healthy at a young age, long-term follow-up studies are necessary, since metabolic alterations could arise later in life and impact the development of adult-onset disorders [221].

## 11. ART and Altered Immune Functions

Several studies reported altered immune functions and higher rates of immune-related diseases in ART-conceived offspring [222,223]. ART-conceived mice responded less efficiently to vaccines and skewed toward T helper (Th) 2-dominated responses [2,224]. Children born after fresh embryo transfer exhibit an increased risk of immune dysfunction in childhood, manifested by elevated Interleukin-4 (IL-4) serum levels and decreased IFN-ɣ/IL-4 ratio [225]. It has also been shown that the expression of three genes, the endoplasmic reticulum aminopeptidase 2 (ERAP2), kynureninase (KYNU), and signal transducer and activator of transcription 4 (STAT4), all of which are involved in immune responses, was altered in placentae of ART-treated women [226,227]. The increased cancer risk in ART conceived individuals might be an indicator of enhanced immune tolerance to tumor antigens [228].

Although the exact etiology is not fully understood, gene–environment interactions appear to play an important role in the development of asthma and allergy [229,230]. ART may enhance the risk of atopic diseases in the offspring through direct epigenetic alterations [231,232]. At the same time, it is important to keep in mind, that ART pregnancies are associated with preterm birth, low birth weight babies, and Cesarean section, all of which are risk factors for asthma [233,234].

Some studies have investigated the risk of atopic disorders in ART-conceived children. A study by Hart et. al. showed that there is no significant increase in asthma and allergies in ART-conceived offspring [235]. Another review by Kettner et. al. concluded that the data on the risk of asthma and allergies among ART conceived are inconsistent [236]. Otherwise, a number of population-based studies as well as systematic reviews and meta-analyses have demonstrated an increased risk of atopy among ART-conceived children [237,238]. A subgroup analysis on singletons and multiple births showed a statistically significant linkage with the risk of asthma only for the former group [238]. However, only four studies were conducted on multiple births, and the results from the larger one were significant [239].

Infertility itself and ART treatment are stressful for couples [240]. Maternal stress or anxiety could increase blood cortisol levels during pregnancy, and subsequent transplacental passage of cortisol could exert a programming effect on the development of the hypothalamic–pituitary–adrenal (HPA) axis in the fetus, resulting in altered stress response later in life [241,242]. In support of these findings, in our previous study, we showed that the prooxidant/antioxidant ratios were slightly higher in IVF-conceived mice than in naturally conceived counterparts, but the difference was not statistically significant [243]. It has been demonstrated that prenatal maternal stress affects asthma development through incomplete development of respiratory tract and offspring immune dysregulation [244]. In addition, evidence suggests that maternal stress may alter the composition of offspring bacterial microbiota. These alterations may adjust the immune system development and enhance the risk of asthma and allergic disease in offspring [245]. ART-associated risks and complications are summarized in Figure 2.

## 12. Conclusions and Future Perspectives

The early stages of embryo development are sensitive to the microenvironment, and even slight changes in the microenvironment could result in long-term consequences for fetal, postnatal, and adult health. ART procedures occur during a critically coordinated period of epigenetic reprogramming, thus the preimplantation period is highly vulnerable to epigenetic aberrations, which could result in cardiometabolic alterations, neurological disorders, increased cancer risk, altered immune responses, asthma, and allergy, as well as other chronic health problems developing in ART-conceived children, adolescents, or adults.

Although there is still a debate on the health status of ART-conceived children, this possibility has to be considered. Investigations on the potential side effects of ART on offspring health status have a long way to go. Long-term follow-up of ART-conceived individuals and transgenerational studies are required. Confounding factors, e.g., infertility itself, the higher maternal age of ART patients, and consequences of multiple embryo transfers, need to be eliminated, given that minor alterations can result in a high risk later in life.

Over time, ART technologies have considerably improved. Using less invasive protocols to induce superovulation should lead to better epigenomes in the oocytes [246], and decreased aneuploidy rates of the resulting embryos [247]. Improved superovulation regimens have been developed for individuals with specific conditions, such as PCOS and cancers [248,249].

The perinatal outcomes of ART have become better, mainly as a result of single embryo transfer and frozen-thawed embryo transfer [250,251,252]. Multiple pregnancy is a risk for prematurity and preterm birth. The goal is to achieve a single pregnancy, by transferring a single embryo. This, however, requires better techniques to identify the competent embryo. Up to now, the selection of gametes and embryos was based on morphological and morphogenetic criteria. Selection based on the morphological features of the embryo is highly prone to subjectivity. Morpho-kinetic measurements provide more objective data. Invasive methods, such as pre-implantation genetic testing for aneuploidy, involve certain risks, since biopsy might negatively influence further development of the embryo. Analysis of spent embryo culture media detects changes that would reflect the metabolism and functional state of the embryo [253,254,255,256]. Further methods include analysis of the cumulus cells using polymerase chain reaction (PCR) and/or DNA microarrays to measure the oocyst’s competency [257,258]. New emerging technologies, such as microfluidic and microfabricated devices, may increase the safety and efficiency of ART procedures [259].

It has been demonstrated that any alteration in either the contact materials (glassware, metalware, plastic ware, etc.) or micro-environment conditions (culture media, freezing media, oxygen, CO_2_, incubator, ambient air, etc.) may exert deleterious effects on the oocyte and resultant embryo quality, consequently enhancing the rate of genetic defects. Therefore, quality assurance and quality control must be considered important steps in decreasing the potential risks associated with ART. Utilization of microfluidic technologies in ART procedures may lead to at least four predictable advantages: (1) exactly controlled gamete or embryo manipulation; (2) establishing a biomimetic environment for culture; (3) promoting microscale genetic and molecular bioassays; and (4) allowing automatization and miniaturization [260].

In spite of the continuously developing technology, the debate on the long-term consequences of ART remains open. Well-controlled epidemiological studies with large sample sizes are necessary. More investigations are also necessary to indicate whether the increased obstetric, perinatal, and health problems observed in ART-conceived children are the direct results of the ART procedure itself or a result of the underlying infertility of the parents. As many parental characteristics cannot be changed, further investigations to identify the optimal ART procedures that improve both perinatal and long-term health status are needed.

## Figures and Tables

**Figure 1 ijms-24-13564-f001:**
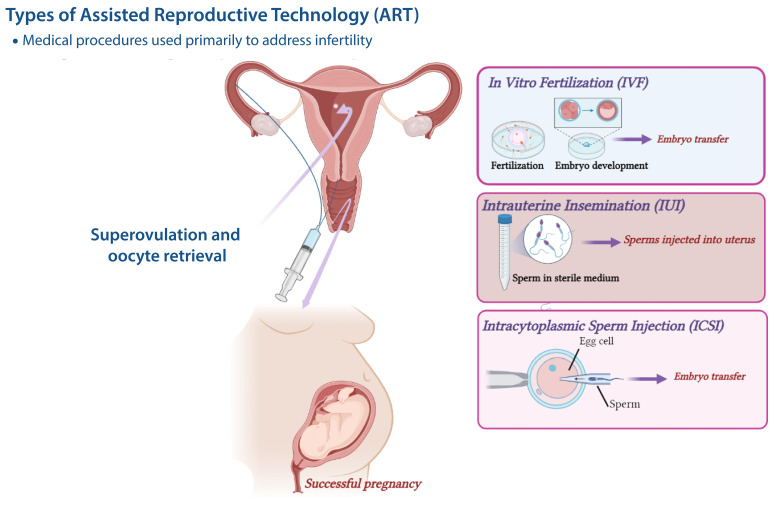
Types of Assisted Reproductive Technology. Common methods of assisted reproductive technology (ART) include: In vitro fertilization (IVF)**,** Intrauterine insemination (IUI), and Intracytoplasmic sperm injection (ICSI). IVF and ICSI involve fertilization outside of the body, the latter with a single sperm injection into a mature egg. IUI involves direct sperm injection into the uterus. A receptive state of the endometrium is *crucial to achieve implantation*. Estrogen and progesterone are essential for establishing a receptive endometrium and successful pregnancy. These hormones are key factors for transition of the endometrium into a receptive state.

**Figure 2 ijms-24-13564-f002:**
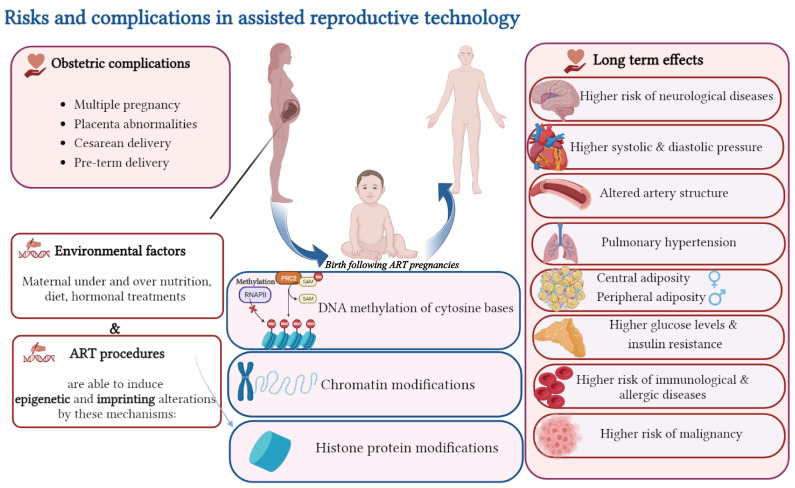
Risks and complications associated with assisted reproductive technologies. ART procedures can induce epigenetic alterations (through three different mechanisms: DNA methylation of cytosine bases, chromatin modifications, and histone protein modifications) and obstetric complications such as placental abnormalities, multiple pregnancy and pre-term delivery. The possible long-term effects of ART include the increased risk of neurological, cardiovascular, metabolic, immunological and allergic diseases.

**Table 1 ijms-24-13564-t001:** Types and effects of preimplantation stress on embryo development and long-term effects.

	Types of Stress	Effects on Embryo Development and Long-Term Effects	Definitions	References
In vitro	IVF	Impaired development Changes to gene expression Altered ICM:TE number and ratio Glucose intolerance	Adding a defined amount of sperm to the oocyte in a culture medium	[40,51]
	ICSI	Impaired development Changes to gene expression Altered ICM:TE number and ratio	A single spermatozoon is injected into the cytoplasm of a mature oocyte.	[52,53]
	Culture media composition	Impaired development Changes to gene expression Modified metabolism Altered ICM:TE number and ratio	Variations in nutrient availability are sources of stresses.	[54,55]
	Oxygen tension	Impaired development Changes to gene expression Altered ICM:TE number and ratio	Oxygen concentration during culture and oxidative stress affect gene expression and intracellular signaling.	[56]
	Temperature	Changes to gene Microtubule disassembly	Temperature fluctuations affect gamete and embryo viability.	[57]
	PH	Impaired development Modified metabolism Changes to gene expression	PH modulates metabolic activity, cellular proliferation, transcriptional activity, protein localization and synthesis.	[36,58]
	Light	Changes to gene expression DNA fragmentation Mitochondrial degradation	White light exposure of the embryo results in impaired implantation capacity.	[45,49,59]
	Substrate stiffness	Different developmental velocity changes to gene expression Altered ICM:TE number and ratio	fertilization and embryo development are more successful on a collagen matrix than on a standard polystyrene petri dish.	[60]
	Cryopreservation	Non-long-term and transgenerational effects	Cryoinjuries including ice crystal formation, structural damage to water bound enzymes, separation of membrane proteins from lipids, altered membrane permeability and osmotic stress due to changes in cell volume should be considered.	[61,62,63]
In vivo	Suboptimal diet	Reduced birthweight Altered growth curves Hypertension	Nutritional conditions in utero are associated with glucose intolerance, obesity, and cardiac dysfunction in adulthood.	[64]
	Maternal diseases	Altered ICM:TE number and ratio	Diabetes, hypertension, epilepsy, obesity, and cardiopulmonary disorders known as in vivo stress factors.	[65,66]
	Endocrine disruptors	Different developmental velocity	Poor oocyte maturation and competency, embryonic defects and poor IVF outcomes are possible complications of endocrine disorders.	[67]
	Toxins	Developmental delay	Tobacco, nicotine can cause delayed migration of embryos from the fallopian tubes into the uterus, growth retardation, pregnancy loss.	[44,68]

## Data Availability

Data availability is not applicable to this article as no new data were created or analysed in this study.

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
