# Peer review of "Long-Term Effects of ART on the Health of the Offspring"

_ijms, 2023, doi:10.3390/ijms241713564_

Round 1

Reviewer 1 Report

The authors are congratulated for developing a comprehensive review of potential risks in offspring created from IVF.  The references are current and while the concerns are raised, balance in the interpretation of data is appreciated. This article will be of benefit to those conducting studies related to longterm effects. The authors highlight deficiencies in the current studies and what will be necessary to improve data collection and analysis. 

Grammatical corrections are included. Please see below. 

2-72 aromatase inhibitors should be added

2-74 administration gonadotropin-releasing hormone (GnRH) agonist and antagonists

The multitude of variations in performing ART complicate the assessment of long-term effects on offspring. Furthermore, the wider application of ART in cancer patients in various stages of diagnosis and treatment create a whole new set of variables that may have long-term effects on offspring.

2-94 “Both clinical and economic evidence favors IUI over IVF in 94 many cases because, during the former, the embryo develops in vivo [18,19]” is stated in a way that could raise appropriate arguments about its validity. This sentence may avoid that concern: “Economic evidence favors an IUI trial of three to four cycles prior to IVF in many cases and provides for in vivo embryo development [18,19].”

2-96,97,98 “In vitro maturation(IVM) of immature oocytes was another important step in ART development [20]. In IVM, immature follicles are not exposed to hormonal stimulation, therefore, this is an alternative approach for IVF, in women with polycystic ovary syndrome (PCOS), who are at risk of ovarian hyperstimulation syndrome.” This is not totally accurate as in many cases of IVM a “gentle” stimulation protocol may be used. Review by a practicing REI specialist will modify the manuscript to accurately reflect current practice nuances.

3-102 “The preimplantation genetic diagnosis was introduced in 1999 and the first live  birth after blastocyst biopsy was reported three years later [22,23].” Preimplantaion genetic testing (PGT) was introduced in……….later [22,23]. PGT adds a number if interventions which may  potentially add to the long-term risks of ART.

3- 118 David Barker was the first to clearly articulate the developmental origin of health and disease theory….

5- 192 Long-Term effects does not need caps.

Table 1 is a useful compilation of literature associated with potential long-term effects. I would check carefully and make sure this table contains all relevant articles.  

6-195,196 … Since animal studies remove infertility as a potentially confounding factor, they allow the assessment of a particular controlled stress and extend investigations with available molecular tools. Such studies may provide useful information that could help to authenticate human ART findings [69,70].

7-246 … group [81].

7-254 … aberrations [88].

7-260 often PGT for aneupolidy, monogenic, structural rearrangement, cryopreservation, thawing, as well as embryo transfer, all of which can potentially disturb normal genomic imprinting [88].

7-262  … the ART procedures ….

7-288 … embryos [103]. Epigenetic disorders….

8-310 … studies [109-111].

10-406 … health risks [141].

11-458, 459 …preimplantation genetic testing (PGT), freezing-thawing, luteal support, and embryo transfer [166-168].

12-515,516 … ART-conceived children [195].

12-518 … reported [195].

12-567 … offspring were challenged….

15-652 … This, however, requires

15-656 … genetic testing for aneuploidy, monogenic and structural rearrangements involve …

Reviewer 2 Report

The authors are dealing with an interesting and important aspect of assisted reproduction and are highlighting causes with the relevant pathomechanism of long-term effects of assisted reproduction. Except for minor issues, the manuscript is well written.

I noticed that the authors are still including GIFT as technique. However, in my years in reproductive medicine, I noticed that this technique is no longer been used. This needs to be corrected, or at least be mentioned in the manuscript. I have added comments and made some corrects in the attached PDF file.

Except for some minor issues, the manuscript is well written.
